# *Indoleamine 2,3-Dioxygenase 2* Deficiency Exacerbates Imiquimod-Induced Psoriasis-Like Skin Inflammation

**DOI:** 10.3390/ijms21155515

**Published:** 2020-08-01

**Authors:** Kento Fujii, Yasuko Yamamoto, Yoko Mizutani, Kuniaki Saito, Mariko Seishima

**Affiliations:** 1Department of Dermatology, Gifu University Graduate School of Medicine, 1-1 Yanagito, Gifu 501-1194, Japan; yoko@gifu-u.ac.jp (Y.M.); seimarik@gifu-u.ac.jp (M.S.); 2Department of Disease Control and Prevention, Fujita Health University Graduate School of Health Sciences, 1-98 Dengakugakubo, Kutsukake, Toyoake, Aichi 470-1192, Japan; yama-y@fujita-hu.ac.jp (Y.Y.); saitok@fujita-hu.ac.jp (K.S.)

**Keywords:** psoriasis, *IDO2*, *IDO1*, tryptophan, kynurenine, IMQ, skin, inflammation, IL-17

## Abstract

*Indoleamine 2,3-dioxygenase 1* (*IDO1*) is an enzyme known to suppress immune responses, and several reports have showed that it is associated with psoriasis. *IDO2* is an isoform of *IDO1*, recently identified as a catalytic enzyme in the tryptophan-kynurenine pathway, which is expressed in dendritic cells and monocytes. The expression of *IDO2* in immune cells suggests that *IDO2* may contribute to immune functions. However, the role of *IDO2* in the pathogenesis of psoriasis remains unclear. In this study, to elucidate the role of *IDO2* in psoriasis, we assessed imiquimod (IMQ)-induced psoriasis-like dermatitis in *IDO2* knockout (KO) mice. Skin inflammation, evaluated by scoring erythema, scaling, and ear thickness, was significantly worse in the *IDO2* KO mice than in the wild-type (WT) mice. The mRNA expression levels of TNF-α, IL-23p19, and IL-17A, key cytokines involved in the development of psoriasis, were also increased in the *IDO2* KO mice. Furthermore, immunohistochemistry revealed that the number of Ki67-positive cells in the epidermis and CD4-, CD8-, and IL-17-positive lymphocytes infiltrating the dermis were significantly increased in the *IDO2* KO mice. These results suggest that *IDO2* might decrease IL-17 expression, thereby resulting in the suppression of skin inflammation in IMQ-induced psoriasis-like dermatitis.

## 1. Introduction

Psoriasis is a chronic inflammatory skin disease that affects 2%–3% of the population [1]. This disease is characterized by the hyperproliferation of keratinocytes and parakeratosis, accompanied by skin tissue infiltration of T cells, monocytes, and neutrophils [2]. The exact etiology of psoriasis remains unknown, but antigenic stimulation causes activation of innate immune cells, which results in the production of proinflammatory cytokines in the skin, leading to the differentiation of Th1 and Th17 cells [2]. Several cytokines, including TNF-α, IL-23, IL-17A, and IL-22, are involved in the pathogenesis of the disease, and neutralizing therapies against TNF-α, IL-12/IL-23-p40, or IL-17A are very efficient for the treatment of patients [3].

Topical treatment with imiquimod (IMQ) induces psoriasiform dermatitis in mice, which shows many features of human psoriasis, including the formation of micro-abscesses, hyperkeratosis, acanthosis, erythema, and scaling [4,5,6]. IMQ is a ligand for toll-like receptor 7/8 [7] and is used in humans for the topical treatment of genital and perianal warts caused by the human papillomavirus, superficial basal cell carcinoma, and actinic keratosis [8,9]. As a side effect, topical treatment with IMQ may cause psoriasis or exacerbate it in patients with well-controlled psoriasis [10,11]. The development of IMQ-induced dermatitis in mice is highly dependent on the IL-23/IL-17 axis [6].

*Indoleamine 2,3-dioxygenase 1* (*IDO1*) is an enzyme that metabolizes tryptophan to kynurenine in the first step of the kynurenine pathway [12]. *IDO1* also acts as an immune modulator in several types of immune cells, including dendritic cells and macrophages [13]. Tryptophan degradation and kynurenine accumulation by *IDO1* induce T cell apoptosis and regulatory T cell proliferation [14,15,16]. In patients with psoriasis, immune cells including dendritic cells and CD4^+^ T lymphocytes have a defect in upregulating *IDO1* in response to inflammation associated with the severity of the disease [17,18].

*Indoleamine 2,3-dioxygenase 2* (*IDO2*) is an isoform of *IDO1* that was recently identified as a catalytic enzyme in the tryptophan-kynurenine pathway [19,20,21]. The genomic structures of these two genes are well conserved, and they are present via duplication on the same chromosome in mammals [19,22,23]. A recent study showed that *IDO1* was expressed in the colon and epididymis, but *IDO2* was expressed mainly in the liver, kidney, and epididymis in mice [19,22,24]. In addition, *IDO2* is expressed in dendritic cells and monocytes and regulates immune functions through autoantibodies and regulatory T cell production [25,26,27,28]. However, the role of *IDO2* in the pathogenesis of psoriasis remains unclear.

In this study, to elucidate the role of *IDO2* in psoriasis, we assessed IMQ-induced psoriasis-like dermatitis in *IDO2* knockout (KO) mice. Based on our data, we propose that *IDO2* might decrease IL-17 expression, leading to the suppression of skin inflammation in IMQ-induced psoriasis-like dermatitis.

## 2. Results

### 2.1. Psoriasiform Lesions in the Imiquimod-Induced Mouse Model of Psoriasis Are Associated with IDO2 But Not with IDO1

To investigate the association of *IDO1* and *IDO2* with psoriasis, we immunohistologically evaluated the normal skin samples of healthy volunteers (*n* = 7), and the lesional and non-lesional skin samples of patients with psoriasis (*n* = 8), using antibodies against *IDO1* and *IDO2*. All skin samples, of both patients with psoriasis and healthy volunteers, were negative for *IDO1* in the epidermis (Figure 1a). Instead, all lesional and non-lesional skin samples of psoriatic patients were positive for *IDO2* in the epidermis. Moreover, while most of the normal skin samples of healthy volunteers were negative for *IDO2*, some showed weak positive signals in the epidermis. 

We next immunohistologically evaluated skin samples from the ears of wild-type (WT) mice treated with vehicle or IMQ for 7 consecutive days. All ear skin samples from vehicle- or IMQ-treated WT mice were negative for *IDO1* in the epidermis (Figure 1b). Although most of the samples from vehicle-treated mice were also negative for *IDO2* in the epidermis, some showed weak positive staining. By contrast, all samples from IMQ-treated mice were positive for *IDO2* in the epidermis. Moreover, we determined the mRNA levels of *IDO1* and *IDO2* in ear skin samples from WT mice treated with vehicle or IMQ for 7 days using quantitative real-time PCR. The mRNA levels of *IDO1* were below the detection sensitivity in WT mice receiving either treatment for 7 days (Figure 1c). Instead, the mRNA levels of *IDO2* were higher in WT mice treated with IMQ than in those treated with vehicle. 

We next compared the effect of IMQ on the skin of *IDO1* KO and WT mice. We applied IMQ cream or Vaseline once daily on both ears of the mice (8–10 weeks of age at the start of treatment) for 7 consecutive days. IMQ treatment induced a psoriasiform skin inflammation in both WT and *IDO1* KO mice. We determined individual scores for erythema, scaling, and ear thickness daily for 7 days. The scores for erythema and scaling did not differ between WT and *IDO1* KO mice treated with IMQ (Figure 1d). Although the ear thickness was higher in IMQ-treated *IDO1* KO mice than WT mice on days 2, 3, and 4 of treatment, there was no difference thereafter. In addition, microscopic observation showed that, after 7 days of treatment, the ear skin epidermis was significantly thicker in WT and *IDO1* KO mice treated with IMQ than in those receiving vehicle (Figure 1e). However, the epidermal thickness did not differ between WT and *IDO1* KO mice receiving IMQ treatment. Thus, these results suggest that psoriasiform lesions in the IMQ-induced mouse model of psoriasis are associated with *IDO2* but not with *IDO1*.

### 2.2. IMQ-Induced Psoriasis-Like Inflammation Is Significantly Increased in IDO2 KO Mice

To elucidate whether IMQ-induced psoriasis-like inflammation is related to *IDO2*, we next assessed the effect of IMQ on the skin of *IDO2* KO and WT mice. We applied IMQ cream or Vaseline once daily on both ears of the mice (8–10 weeks of age at the start of treatment) for 7 consecutive days. IMQ treatment induced a psoriasiform skin inflammation in both WT and *IDO2* KO mice. However, the deterioration of psoriasis-like skin lesions on the ear after 7 days of IMQ treatment was more pronounced in *IDO2* KO than in WT mice (Figure 2a). Individual scores for erythema, scaling, and ear thickness were significantly increased in *IDO2* KO mice. In particular, all scores on days 5 and 7 after topical IMQ were significantly higher in *IDO2* KO than in WT mice (Figure 2b). Scores in vehicle-treated mice remained normal (Figure 2a,b). Microscopic evaluation of ear sections from WT and *IDO2* KO mice after 7 days of IMQ treatment revealed characteristic changes in psoriasiform lesions, including hyperkeratosis, parakeratosis, acanthosis, and micro-abscess formation, which were not observed in vehicle-treated mice. In addition, the IMQ-induced skin lesions in *IDO2* KO mice showed more characteristic changes than those in WT mice (Figure 2c). Epidermal thickness of the ear skin from WT and *IDO2* KO mice, microscopically measured after 7 days of IMQ treatment, was significantly increased. Moreover, the epidermis of the ears of *IDO2* KO mice treated with IMQ was significantly thicker than that of WT mice (Figure 2d), indicating that IMQ-induced psoriasis-like skin inflammation was significantly exacerbated in *IDO2* KO mice.

### 2.3. The Deletion of IDO2 Affects the mRNA Expression of Key Cytokines Involved in the Pathogenesis of Psoriasis

We determined the mRNA expression levels of several inflammatory cytokines that are important in psoriasis (TNF-α, IL-23p19, IL-17A, IL-22, IFN-α, IL-1β, IL-6, and IL-10) and a chemokine that promotes the migration of neutrophils (CXCL2) using quantitative real-time PCR [2,29,30]. Since the peak-time of activity is different for each cytokine, we measured the mRNA levels of cytokines in the ear skin of WT and *IDO2* KO mice before and 24 h, 48 h, and 7 days after IMQ treatment (Figure 3). Cytokine mRNA levels were not significantly different between WT and *IDO2* KO mice before and 24 h after IMQ treatment. However, 48 h and 7 days after IMQ treatment, the mRNA levels of most key cytokines involved in psoriasis development were higher in *IDO2* KO than in WT mice. Furthermore, the expression of TNF-α, IL-23p19, and IL-17A was significantly higher in *IDO2* KO than in WT mice on day 7. Thus, these results show that the deletion of *IDO2* affects the mRNA expression of key cytokines for the pathogenesis of psoriasis.

### 2.4. Keratinocyte Proliferation and Inflammatory Cell Infiltration Are Significantly Induced in IMQ-Treated IDO2 KO Mice

We next examined the immunohistological differences between WT and *IDO2* KO mice treated with IMQ or vehicle after 7 days. Firstly, to assess keratinocyte proliferation in the ear skin of mice, we stained for Ki-67, a cell proliferation marker (Figure 4a,b). We counted the number of Ki-67-positive cells in the epidermis per high power field. Treatment with IMQ significantly increased the number of Ki-67-positive cells in WT and *IDO2* KO mice. In addition, the number of Ki-67-positive cells was significantly higher in IMQ-treated *IDO2* KO mice than WT mice. Secondly, to test T cell infiltration in the dermis, we investigated the expression of CD4 and CD8, which are the markers of helper and cytotoxic T cells, respectively (Figure 4a,b). We counted the number of CD4- or CD8-positive cells in the dermis per high power field. IMQ treatment significantly increased the number of CD4- or CD8-positive cells in WT and *IDO2* KO mice. The number was significantly higher in IMQ-treated *IDO2* KO than WT mice. Finally, because IL-17 signaling plays an important role in psoriasis [6,31,32], we counted the number of IL-17-positive cells in the dermis per high power field (Figure 4a,b). Similarly, the number of IL-17-positive cells in WT and *IDO2* KO mice receiving IMQ treatment was significantly higher compared to mice receiving vehicle. Moreover, we found a significantly greater number of IL-17-positive cells in IMQ-treated *IDO2* KO mice than WT mice. Therefore, these results show that keratinocyte proliferation and infiltration of inflammatory cells were significantly induced in *IDO2* KO mice treated with IMQ, and IMQ-induced psoriasis-like dermatitis is induced in *IDO2* KO mice via IL-17-positive lymphocytes.

### 2.5. Exacerbation of IMQ-Induced Psoriasis-Like Skin Inflammation in IDO2 KO Mice Is Associated with IDO2 Itself rather than Tryptophan Metabolites

To elucidate the mechanism by which *IDO2* deficiency exacerbates IMQ-induced psoriasis-like skin inflammation, we next measured the concentration of tryptophan metabolites, including tryptophan, kynurenine, kynurenic acid, anthranilic acid, and 3-hydroxy anthranilic acid (3-HAA), in the ears of WT and *IDO2* KO mice treated with vehicle or IMQ for 7 days, using high performance liquid chromatography (HPLC). Although it is generally considered that the concentration of kynurenine decreases upon deletion of *IDO2* gene, unexpectedly, it did not decrease in *IDO2* KO mice compared to WT mice. Moreover, tryptophan metabolites (kynurenine, kynurenic acid and anthranilic acid) were found in higher concentrations in IMQ-treated *IDO2* KO mice than WT mice (Figure 5). Kynurenine is known to actively suppress the immune system by binding to the aryl hydrocarbon receptor, thereby inducing regulatory T cells [33,34]. Kynurenic acid also suppresses the production of IL-17 and IL-23 in CD4^+^ T cells and dendritic cells in vitro [35]. Thus, the accumulation of kynurenine and kynurenic acid is associated with severe inflammation, resulting in an increase in the downstream metabolites of kynurenine, kynurenic acid, and anthranilic acid. These results show that exacerbation of IMQ-induced psoriasis-like skin inflammation in *IDO2* KO mice might be associated with *IDO2* itself rather than with tryptophan metabolites.

## 3. Discussion

Our results showed that *IDO2* deletion in mice exacerbated IMQ-induced psoriasis-like dermatitis. IL-17 signaling plays an important role in IMQ-induced dermatitis models as well as psoriasis [6,31,32]. Interestingly, we found a remarkably higher mRNA level of IL-17A and more IL-17-positive cells infiltrating the dermis of *IDO2* KO mice treated with IMQ than in WT mice. This suggests that *IDO2* acts on the IL-17 signaling pathway and consequently suppresses IL-17 production.

The relationship between the pathophysiology of psoriasis and *IDO1* has previously been reported. Although *IDO1* has a protective effect in psoriasis by inducing regulatory T cells, immune cells from patients with psoriasis have a defect in upregulating *IDO1* in response to inflammation which is proportional to the disease severity [17,18]. However, our data show that the IMQ-induced mouse model of psoriasis was not associated with *IDO1*. One reason might be that the IMQ-induced mouse model does not accurately recapitulate human psoriasis because of the lack of chronicity or comorbidities of psoriasis [36]. Instead, the role of *IDO2* in psoriasis had remained unclear until now. Our data show that *IDO2* might suppress IMQ-induced psoriasis-like skin inflammation. With respect to *IDO1* and skin, keratinocytes and fibroblasts can upregulate *IDO1* expression and activation in vitro in response to proinflammatory stimuli [37,38]. Nevertheless, myeloid dendritic cells are the major *IDO1*-expressing skin cells in vivo, and *IDO1* is not expressed in the epidermis [39]. Our results also show that, in the epidermis of patients with psoriasis and WT mice treated with IMQ, *IDO2* was expressed but *IDO1* was not. In addition, while the deletion of IL-17RA in T cells, neutrophils, or macrophages has no impact on IMQ-induced dermatitis, only deletion of this receptor in keratinocytes reflects the effects of its systemic deletion, resulting in strongly reduced dermatitis development [32]. Therefore, because *IDO1* is not expressed or weakly expressed in the epidermis, it might not affect IMQ-induced dermatitis. By contrast, because *IDO2* is strongly expressed in the epidermis, it might influence IMQ-induced dermatitis. In future experiments, it will be necessary to selectively remove cells such as keratinocytes, dendritic cells, and monocytes in order to confirm this hypothesis.

The mechanism by which *IDO2* controls IMQ-induced dermatitis remains unclear. In our previous report, *IDO2* was demonstrated to be important in the regulation of STAT3 signaling using a lipopolysaccharide-induced endotoxin shock model [40]. STAT3 participates in signaling downstream of multiple cytokines implicated in psoriasis, such as IL-6, IL-10, IL-20, IL-22, and IL-23, and may have a role in mediating the innate immune response in psoriatic epidermis [41]. STAT3 is also required for the differentiation of Th17 cells [42]. Therefore, *IDO2* might modulate STAT3 signaling, leading to decreased IL-17 expression and suppressing IMQ-induced dermatitis.

To investigate whether the mechanism that regulates IMQ-induced dermatitis by *IDO2* is associated with tryptophan metabolites, we measured the concentration of tryptophan metabolites in the ears of treated mice using HPLC. Although the concentration of kynurenine is generally expected to decrease when *IDO2* is deleted, in this study, it was significantly higher in IMQ-treated *IDO2* KO mice than in WT mice. Previous reports have shown that *IDO2* has a low affinity for tryptophan in a cell-free assay, resulting in a lower catalytic efficiency compared with *IDO1* [43,44,45]. Therefore, the mechanism by which *IDO2* regulates IMQ-induced dermatitis might not be associated with tryptophan metabolites, and our results on the metabolites might just reflect the consequences of inflammation caused by IMQ treatment.

## 4. Materials and Methods

### 4.1. Mice

All mice used in experiments were 8–10-week-old females. *IDO2* KO mice on a C57BL/6N background were obtained from the Knockout Mouse Project (KOMP, CA, USA). Mice that were homozygous null (*IDO2* KO) by targeted disruption of the *IDO2* gene were selected from the offspring of heterozygous-homozygous mating based on PCR of tail DNA for genotyping. *IDO1* KO mice on a C57BL/6 background were obtained from Jackson Laboratory (Bar Harbor, ME, USA). We purchased WT C57BL/6N mice from Charles River Laboratories (Yokohama, Japan). Mice were housed in the animal facilities of Fujita Health University Graduate School of Medicine under specific pathogen-free conditions, maintained at 25 °C on a standard 12-h light/dark cycle (lights on at 08:00) and with free access to food and water. The protocol for all animal experiments was approved by the Animal Experimentation Committee of Fujita Health University Graduate School of Medicine (AP19067, approved on 20 May 2019). Procedures involving mice and their care conformed to international guidelines, as described in the Principles of Laboratory Animal Care (National Institutes of Health publication 85–23, revised in 1985).

### 4.2. IMQ-Induced Psoriasis-Like Inflammation Models

WT and IDO2 KO mice at 8–10 weeks of age received a daily topical dose of 62.5 mg commercially available IMQ cream (5%) (Mochida Pharmaceutical, Tokyo, Japan) on both ears for 7 consecutive days. Vaseline (Maruishi Pharmaceutical, Osaka, Japan) was used as a vehicle control. To evaluate the severity of inflammation of the ear skin, an objective scoring system was assessed based on the clinical psoriasis area and severity index (PASI), except for the affected area. The ear thickness was measured in millimeters, and erythema and scaling were scored independently on a scale from 0 to 4: 0, none; 1, slight; 2, moderate; 3, marked; 4, very marked. 

### 4.3. Patients

This study was approved by the ethical committee of Gifu University and was performed in strict adherence with the Declaration of Helsinki principles (2020-087, approved on 01 Jul 2020). Skin biopsy specimens were taken from healthy volunteers and patients with psoriasis that were diagnosed clinically and histopathologically in our department.

### 4.4. RNA Extraction and Quantitative Real-Time PCR

Total RNA was extracted from ear tissue using Isogen Ⅱ (Nippon Gene, Tokyo, Japan) and reverse transcription-PCR was carried out using ReverTra Ace qPCR RT kits (Toyobo, Osaka, Japan). Quantitative real-time PCR was performed using SYBR Green (SsoAdvanced Universal SYBR Green SuperMix; Bio-Rad, Hercules, CA, USA) with an ABI PRISM-7900HT (Applied Biosystems, Foster City, CA, USA). The mRNA expression of target genes was normalized to Glyceraldehyde-3-phosphate dehydrogenase (GAPDH). The primer sequences were as follows: GAPDH, sense 5′-CAATGTGTCCGTCGTGGATCT-3′, antisense 5′-GTCCTCAGTCTAGCCCAAGATG-3′; IL-17A, sense 5′-AGCTGGACCACCACATGAATTC-3′, antisense 5′-AGCATCTTCTCGACCCTGAAAG-3′; IL-22, sense 5′-GGTGACGACCAGAACATCA-3′, antisense 5′-CAGCAGGTCCAGTTCCCCAAT-3′; IL-23p19, sense 5′-AATAATGTGCCCC GTATCCA-3′, antisense 5′-CATGGGGCTATCAGGGAGTA-3′; IFN-α, sense 5′-CCTGCTGGCTGTGAGGACAT-3′, antisense 5′-GGCTGTGTTTCTTCTTTCTCAGGTA-3′; IL-1β, sense 5′-TCCACCTCAATGGACAGAATATCA-3′, antisense 5′-GGTTCTCCTTGTACAAAGCTCATG-3′; IL-6, sense 5′-ATGGATGCTACCAAACTGGAT-3′, antisense 5′-TGAAGGACTCTGGCTTTGTCT-3′; IL-10, sense 5′-CTGGCTCAGCACTGCTATGC-3′, antisense 5′-CTGGCCCCTGCTGATCCT-3′; TNF-α, sense 5′-GCCAGCCGATGGGTTGT-3′, antisense 5′-GCAGCCTTGTCCCTTGAAGA-3′; CXCL2, sense 5′-TTGCCTTGACCCTGAAGCCCCC-3′, antisense 5′-GGCACATCAGGTACGATC CAGGC-3′; IDO1, sense 5′-AGTTGGGCCTGCCTCCTATTC-3′, antisense 5′-GAAGAAGCCCTTG TCGCAGTC-3′; IDO2, sense 5′-CATACCAGGCAATTGCTCCAC3′, antisense 5′-GCCTGGGCTA AAGAGCTCAATAC-3′.

### 4.5. Histology and Immunohistochemistry

Paraffin-embedded specimens were prepared using standard methods. Sections were stained with hematoxylin and eosin (H & E) and immunohistochemical staining. The primary Abs were anti-IDO1 (ab106134; Abcam, Cambridge, UK), anti-IDO2 (ab214214; Abcam), anti-CD3 (ab5690; Abcam), anti-CD4 (ab183685; Abcam), anti-CD8 (ab203035; Abcam), anti-IL-17 (ab79056; Abcam), and anti-Ki-67 (NB500-170; Novus Biologicals, Centennial, USA). Histofine Simple Stain MAX PO (Nichirei Biosciences, Tokyo, Japan) was used for mouse samples and Dako Envision System-Labeled Polymer HRP (Dako, Tokyo, Japan) for human samples as secondary antibodies. 

### 4.6. Measurement of Tryptophan Metabolites

Tryptophan, kynurenine, kynurenic acid, anthranilic acid, and 3-HAA were measured by HPLC (SHIMAZU, Kyoto, Japan). Ear tissue was weighed and homogenized (1:5, *w*/*v*) in 10% perchloric acid. The mixture was centrifuged at 13,000 rpm for 15 min (4 °C). Thereafter, 50 μL of the resulting supernatant was injected into the HPLC instrument for quantification. Tryptophan, kynurenine, kynurenic acid, anthranilic acid, and 3-HAA were eluted from a reverse-phase column by an isocratic method (TSKgel ODS-100 mv 3 μm 4.6 mm [ID] × 15 cm [L]), using a mobile phase with 10 mmol/L sodium acetate and 1.7−2.0% acetonitrile (pH adjusted to 4.5 with acetic acid) at a flow rate of 0.8 mL/min. Tryptophan and kynurenine were measured by a photodiode array detector (SPD-M30A: SHIMAZU, Kyoto, Japan) at wavelengths of 280 nm and 365 nm, respectively. Kynurenic acid, anthranilic acid, and 3-HAA were measured by a fluorescence detector (RF-20Axs: SHIMAZU, Kyoto, Japan) at excitation/emission wavelengths of 320 nm and 420 nm for anthranilic acid and 3-HAA and of 334 nm and 380 nm for kynurenic acid, respectively.

### 4.7. Statistical Analysis

All data are expressed as the mean ± SEM. Statistically significant differences between groups were determined using Student’s *t*-test, one-way ANOVA, or two-way ANOVA. A *p* value of less than 0.05 was considered statistically significant. GraphPad Prism software version 6 was used for all statistical analyses.

## 5. Conclusions

Our results suggest that IDO2 might modulate the IL-17 signaling pathway, leading to decreased IL-17 expression and suppressing the skin inflammation in IMQ-induced psoriasis-like dermatitis. In addition, IDO2 may provide an effective means of treating psoriasis.

## Figures and Tables

**Figure 1 ijms-21-05515-f001:**
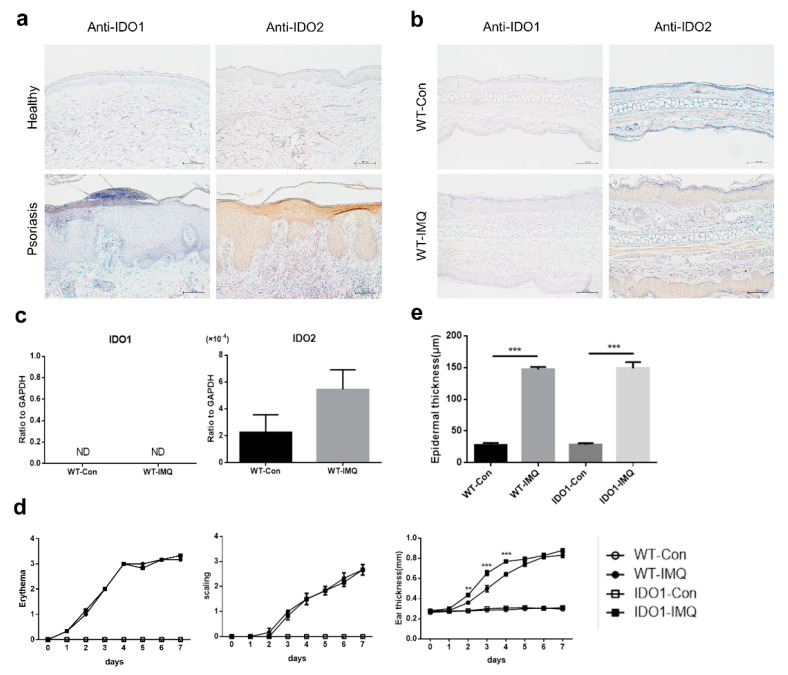
Psoriasiform lesions in the imiquimod-induced mouse model of psoriasis are associated with *IDO2* but not with *IDO1*. (**a**) Immunohistochemical staining of the skin from healthy volunteers and patients with psoriasis using antibodies to *IDO1* and *IDO2*. Scale bar = 100 µm. (**b**) Immunohistochemical staining of the ear from wild-type (WT) mice treated with vehicle or imiquimod (IMQ) for 7 days using antibodies to *IDO1* and *IDO2*. Scale bar = 100 µm. (**c**) The mRNA levels of *IDO1* and *IDO2* in the ears of WT mice treated with vehicle or IMQ for 7 days, detected using quantitative real-time PCR. Data are presented as mean ± SEM. (Student’s *t*-test). Control group, *n* = 6; IMQ-treated group, *n* = 9. (**d**) Erythema, scaling, and thickness of the ear in WT and *IDO1* KO mice treated with vehicle or IMQ for 7 days were evaluated daily (0 = none, 1 = slight, 2 = moderate, 3 = marked, 4 = very marked). ** *p* < 0.01, *** *p* < 0.001 when comparing *IDO1* KO-IMQ with WT-IMQ (two-way ANOVA). Control groups, *n* = 3; IMQ-treated groups, *n* = 6. (**e**) Microscopic evaluation of epidermal hyperplasia of the ear in WT and *IDO1* KO mice treated with vehicle or IMQ for 7 days. Data are presented as mean ± SEM. *** *p* < 0.001 (one-way ANOVA). Control groups, *n* = 3; IMQ-treated groups, *n* = 6. ND: not-detected, WT: wild type, *IDO*: *indoleamine 2,3-dioxygenase*, Con: control, IMQ: imiquimod, KO: knockout.

**Figure 2 ijms-21-05515-f002:**
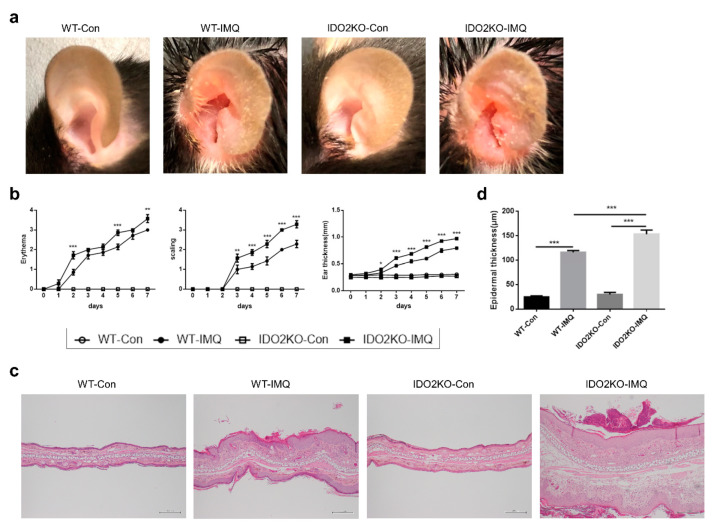
IMQ-induced psoriasis-like inflammation is significantly more severe in *IDO2* KO mice. (**a**) Macroscopic phenotypic representation of the left ear in WT and *IDO2* KO mice treated with vehicle or IMQ for 7 days. (**b**) Erythema, scaling, and thickness of the ear were evaluated daily (0 = none, 1 = slight, 2 = moderate, 3 = marked, 4 = very marked). Data are representative of three independent experiments. * *p* < 0.05, ** *p* < 0.01, *** *p* < 0.001 when comparing *IDO2* KO-IMQ to WT-IMQ (two-way ANOVA). Control groups, *n* = 4; IMQ-treated groups, *n* = 7. (**c**) Hematoxylin and eosin staining of the ear in WT and *IDO2* KO mice treated with vehicle or IMQ for 7 days. Scale bar = 200 µm. (**d**) Microscopic evaluation of epidermal hyperplasia after 7 days of treatment. Data are presented as mean ± SEM. *** *p* < 0.001 (one-way ANOVA). Control groups, *n* = 6–7; IMQ-treated groups, *n* = 9. WT: wild type, *IDO*: *indoleamine 2,3-dioxygenase*, Con: control, IMQ: imiquimod, KO: knockout.

**Figure 3 ijms-21-05515-f003:**
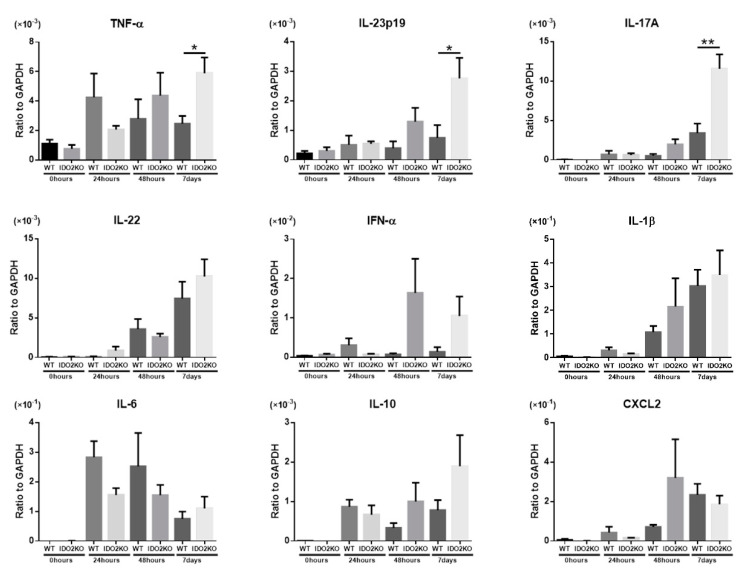
The deletion of *IDO2* affects the mRNA expression of key cytokines involved in the pathogenesis of psoriasis. Quantitative real-time PCR analysis of the ear before and 24 h, 48 h, and 7 days after IMQ treatment. Data are presented as mean ± SEM. * *p* < 0.05, ** *p* < 0.01 (Student’s *t*-test). *n* = 4–7. WT: wild type, *IDO*: *indoleamine 2,3-dioxygenase*, IMQ: imiquimod, KO: knockout.

**Figure 4 ijms-21-05515-f004:**
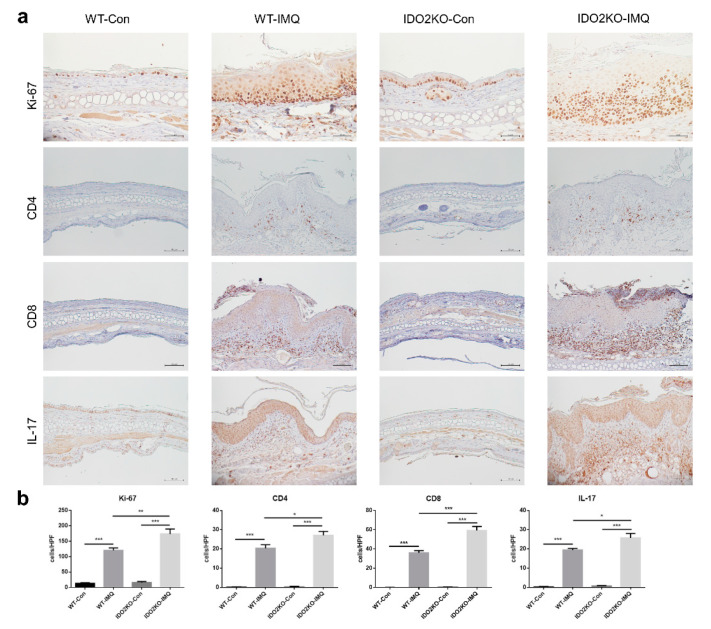
Keratinocyte proliferation and inflammatory cell infiltration are significantly increased in *IDO2* KO mice treated with IMQ. (**a**) Immunohistochemical staining of the ear of WT and *IDO2* KO mice treated with vehicle or IMQ for 7 days using Ki-67, CD4, CD8, and IL-17 antibody. Ki-67 antibody, scale bar = 50 µm; CD4, CD8, and IL-17 antibody, scale bar = 100 µm. (**b**) The number of Ki67^+^ cells in the epidermis and infiltrating CD4^+^, CD8^+^, and IL-17^+^ lymphocytes in the dermis per high power field. Data are presented as mean ± SEM. * *p* < 0.05, ** *p* < 0.01, *** *p* < 0.001 (one-way ANOVA). Control groups, *n* = 6–7; IMQ-treated groups, *n* = 9. WT: wild type, *IDO*: *indoleamine 2,3-dioxygenase*, Con: control, IMQ: imiquimod, KO: knockout.

**Figure 5 ijms-21-05515-f005:**
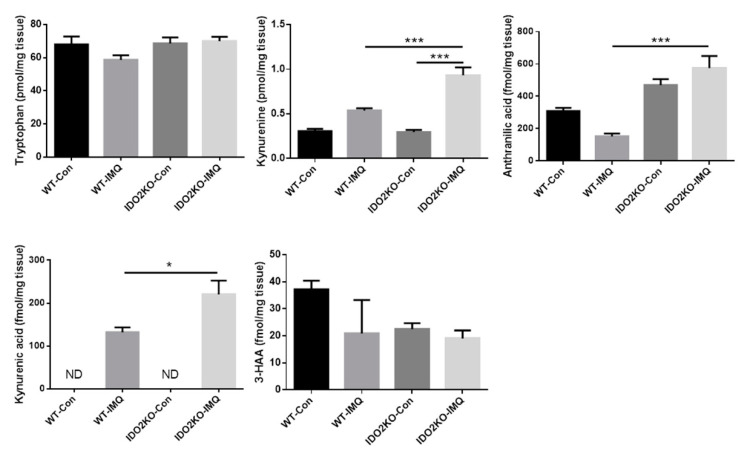
The concentration of tryptophan metabolites is higher in IMQ-treated *IDO2* KO mice than WT mice. The concentration of tryptophan, kynurenine, kynurenic acid, anthranilic acid, and 3-HAA in the ears of WT and *IDO2* KO mice with vehicle or IMQ treatment for 7 days were measured by high performance liquid chromatography. Data are presented as mean ± SEM. * *p* < 0.05, *** *p* < 0.001 (one-way ANOVA). Control groups, *n* = 4; IMQ-treated groups, *n* = 7. 3-HAA: 3-hydroxy anthranilic acid, ND: not-detected, WT: wild type, *IDO*: *indoleamine 2,3-dioxygenase*, Con: control, IMQ: imiquimod, KO: knockout.

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
