# Peer review of "Indoleamine 2,3-Dioxygenase 2 Deficiency Exacerbates Imiquimod-Induced Psoriasis-Like Skin Inflammation"

_ijms, 2020, doi:10.3390/ijms21155515_

Round 1

Reviewer 1 Report

This manuscript presents results of studies on the role of 2,3-dioxygenase 2, an isoform of the indoleamine 2,3-dioxygenase 1 in psorasis or more general psorasis-like disease. The research design is right, the methods are adequately chosen and the manuscript is scientifically sound. The only question to be answered is whether the fact that an isoform of an enzyme with known properties, display more or less the same activity is worth publishing as a full-length paper in International Journal of Molecular Sciences.

Author Response

We would like to thank you for your careful reading of our manuscript and for your helpful comments.

Comment: This manuscript presents results of studies on the role of 2,3-dioxygenase 2, an isoform of the indoleamine 2,3-dioxygenase 1 in psorasis or more general psorasis-like disease. The research design is right, the methods are adequately chosen and the manuscript is scientifically sound. The only question to be answered is whether the fact that an isoform of an enzyme with known properties, display more or less the same activity is worth publishing as a full-length paper in International Journal of Molecular Sciences.

Response: Thank you for your suggestion.It is becoming clear that IDO1 and IDO2 have different localizations and different mechanisms of action.
In addition, the relationship between IDO1 and psoriasis is gradually becoming clear, but the relationship between IDO2 and psoriasis is not clear at all.
Furthermore, our result that IDO1 has almost no effect on psoriasis-like dermatitis, while IDO2 affects it may further contribute to the elucidation of the mechanism of psoriasis and be useful for the treatment of psoriasis. Therefore, We believe that our paper is worth publishing as a full-length paper in International Journal of Molecular Sciences.

Reviewer 2 Report

The manuscript intituled ¨Indoleamine 2,3-dioxygenase 2 deficiency exacerbates imiquimod-induced psoriasis-like skin inflammation¨is well written and showed a good design. The authors showed the role of IDO2 in psoriasis, after imiquimod (IMQ)-induced psoriasis-like dermatitis in IDO2 knockout (KO) mice. I recommend some clarifications before the final decision.

  1. Please, English needs to revise.
  2. The authors need to improve the resolution of all figures, also Immunohistochemical staining images need to be improved. Is not possible to identify well each one.
  3. It should be interesting if the authors include a conclusion section.
  4. The authors concluded that ¨In summary, our results suggest that IDO2 might result in decreased IL-17 expression and suppress the skin inflammation in IMQ-induced psoriasis-like dermatitis. Please justify better this sentence in the manuscript.

Author Response

We would like to thank you for your careful reading of our manuscript and for your helpful comments.

Comment: 1.Please, English needs to revise.

Response: Thank you for your suggestion. We have asked Editage to edit the English, so we attached the certificate.

Comment:2.The authors need to improve the resolution of all figures, also Immunohistochemical staining images need to be improved. Is not possible to identify well each one.

Response: Thank you for your suggestion. We have improved all figures.

Comment:3.It should be interesting if the authors include a conclusion section.

Response: Thank you for your suggestion. Accordingly, we have included a conclusion section.

Comment:4. The authors concluded that ¨In summary, our results suggest that IDO2 might result in decreased IL-17 expression and suppress the skin inflammation in IMQ-induced psoriasis-like dermatitis. Please justify better this sentence in the manuscript.

Response: Thank you for your suggestion. Accordingly, we have changed it to "Our results suggest that IDO2 might modulate the IL-17 signaling pathway, leading to decreased IL-17 expression and suppressing the skin inflammation in IMQ-induced psoriasis-like dermatitis."
